# Embedded Processing and Compression of 3D Sensor Data for Large Scale Industrial Environments

**DOI:** 10.3390/s19030636

**Published:** 2019-02-02

**Authors:** Joacim Dybedal, Atle Aalerud, Geir Hovland

**Affiliations:** Department of Engineering Sciences, University of Agder, 4879 Grimstad, Norway; atle.aalerud@uia.no (A.A.); geir.hovland@uia.no (G.H.)

**Keywords:** 3D sensors, point clouds, time-of-flight, lidar, compression, denoising, scalability

## Abstract

This paper presents a scalable embedded solution for processing and transferring 3D point cloud data. Sensors based on the time-of-flight principle generate data which are processed on a local embedded computer and compressed using an octree-based scheme. The compressed data is transferred to a central node where the individual point clouds from several nodes are decompressed and filtered based on a novel method for generating intensity values for sensors which do not natively produce such a value. The paper presents experimental results from a relatively large industrial robot cell with an approximate size of 10 m × 10 m × 4 m. The main advantage of processing point cloud data locally on the nodes is scalability. The proposed solution could, with a dedicated Gigabit Ethernet local network, be scaled up to approximately 440 sensor nodes, only limited by the processing power of the central node that is receiving the compressed data from the local nodes. A compression ratio of 40.5 was obtained when compressing a point cloud stream from a single Microsoft Kinect V2 sensor using an octree resolution of 4 cm.

## 1. Introduction

The rapid development of 3D sensors based on the time-of-flight principle is currently an important enabling factor for different autonomous systems, such as self-driving vehicles or drones. However, there is also an increasing interest in using these types of sensors in other industries where the sensors are not moving with a vehicle or machine, but are mounted in fixed locations and monitoring a volume of interest. One such application is within the offshore oil and gas industry, where, in recent years, there has been a growing focus on automation, safety systems and better efficiency (see, for example, [1]). This is highly motivated by cost reductions due to lower investment activity by exploration and production companies. Adding depth cameras, lidar (Light Detection And Ranging) and other 3D sensors to existing and newly built facilities could contribute to enabling and accelerating this development.

One feature of the 3D sensors is the generation of a potentially large amount of data, requiring a high bandwidth between the sensor and the computer processing the data. For example, the Microsoft Kinect for Xbox One (Kinect V2) sensor (Microsoft Corporation, Redmond, WA, USA) can generate data of several Gbit/s on a USB3 connection. With a dedicated PCI card, it is possible to use a few such sensors on a single computer; however, scaling the solution up to many more sensors is difficult with a centralized solution. A first step to ensure scalability is to lower the bandwidth requirement using point cloud compression. The following subsection will describe previous methods for compressing point cloud data and the main contributions of this paper will be outlined in Section 1.2.

### 1.1. Related Work

In [2], a compression and decompression scheme for point cloud streams was presented. The paper presented a novel XOR-based differential octree used by the compressor to detect temporal changes between two consecutively measured point clouds. The authors in [2] implemented both spatial and temporal compression of arbitrary point clouds originating from a single Kinect depth sensor. It was shown that octrees can be used to efficiently compress and encode such point cloud streams. The authors also showed that the double-buffered octree compression scheme was suitable for compressing even unstructured point clouds. Temporal compression was described to be especially effective when there is a limited amount of movement in the point cloud, which is the case in our application where most of the scenery is static and movement only comes from the machines and/or humans. In addition to the implicit grid filtering performed by inserting points into an octree with a fixed minimum voxel size, the points in the original point cloud could be reconstructed with a given precision at the decompressor due to an additional encoding of point details in each octree leaf node.

The underlying compression algorithm used in [2] is a variant of an arithmetic entropy encoder known as an integer-arithmetic or range encoder originally presented in [3]. In short, a range encoder utilizes the probability distribution of a set of symbols (e.g., bytes in this case) with a given range, and encodes the symbols into one number by dividing the initial range into sub-ranges based on the symbol probabilities. The reader is referred to [3] for further details. The range encoder implemented by [2] takes a stream of bytes as an input, and produces an encoded stream which can be stored or transmitted over a network.

A dynamic compression scheme based on [2] was developed in [4]. Using a single Kinect depth sensor, the authors were able to achieve an average frame rate of 5.86 Hz on a network subjected to different levels of background transmissions. In our scenario, such background noise on the network is minuscule, as the entire network is designated to transfer data between the sensors, the central computer and the machines in the lab.

Another technique that has been shown to be useful for processing point clouds is graph signal processing (GSP). In [5,6], GSP was used to resample and compress point clouds, respectively. Resampling was achieved by applying different filters on the generated graphs and compression was partially based on the double-buffered octree scheme outlined above. Furthermore, in [7], GSP was used to perform denoising by first removing outlier points based on a weighted graph (similar to a statistical outlier removal filter), and subsequently smoothing the point cloud by means of convex optimization on the graph signal. While these results show that GSP can be well suited for working with point clouds, they also show that the process can be time-consuming. For example, in [5], the authors were able to process 15 million 3D points in 1000 s, equivalent to 15,000 points per second. However, since the Kinect V2 used in this paper generates point clouds consisting of 217,088 points at rates up to 30 Hz, it would require a much lower processing time to keep up with such frame rates.

In [8], an octree based mapping framework (OctoMap) was proposed. The generated map was implemented as an occupancy grid, where voxels are labeled occupied, free or unknown based an occupancy probability. The work included a compression scheme combining clamping and octree pruning, where the tree is pruned when all children of a tree node are considered stable (the nodes have an occupancy probability close to 0 or 1). The framework, which also incorporates probabilistic sensor fusion, is intended to build a map of an environment, and thus is of interest in our scenario where the goal is to create a dynamic map of a robotic cell. If all sensors were directly connected to the central computer, OctoMap could be used to fuse the sensor data and generate a map of the complete area. However, the focus of this paper is to lower the amount of data generated by the sensors before the data is inserted into such a map, limiting the required network bandwidth between the sensors and the central computer, and ensuring scalability.

### 1.2. Main Contributions

Unlike in most literature where single sensors often are considered [4,9,10], an industrial application will typically need multiple sensors to cover relatively large areas. To ensure scalability, a new approach is presented, where each sensor is connected to a local computer. The generated point cloud data is processed and compressed on the embedded sensor node before the data is sent to a central computer for decompression and further processing.

To perform the compression, the method presented in [2] has been implemented on embedded hardware. The method has further been modified by introducing an attribute that represents the “intensity” of each voxel. The modification was made in order to allow filtering of noisy measurements (outliers) at the central computer and to introduce a trust level indicator for each voxel. Implementation of such an intensity value for RGB-D (color and detph) sensors was also motivated by intensity values returned by lidar sensors and the potential fusion of data from different sensor types.

A compression ratio of 40.5 is achieved, and a denoising scheme based on the calculated point intensities is proposed. With such a setup, the scalability problem is solved by decentralization.

## 2. Materials and Methods

This paper describes a processing, compression and transmission framework for point cloud data generated by 3D sensors. The software is developed as two Robot Operating System (ROS, [11]) nodes. The first ROS node is deployed on an NVIDIA Jetson TX2 module (NVIDIA Corporation, Santa Clara, CA, USA) which, in addition to a CPU, contains a general purpose graphical processing unit (GPGPU). The module may have one or more 3D sensors connected, and together they form a sensor node. By exploiting the processing capabilities of the Jetson TX2, data from the connected sensor(s) is processed and compressed locally before it is published on the ROS network. The second ROS node is deployed on a central computer and receives the compressed data from one or more sensor nodes. In the following subsections, the different processing steps are described. The developed software is available in public github repositories, see [12].

### 2.1. Problem Formulation and Motivation

The sensor data from multiple time-of-flight 3D sensors is to be used as input to a “GPU Voxels”-based application [13], a GPU based collision detection software which is running on a centralized computer. Similar to OctoMap, a voxel-based occupancy grid is created and, due to the fact that the map is stored in the GPU memory, calculations such as distance to the nearest object can be performed efficiently on multiple voxels in parallel and in real-time. As the application requires a fixed voxel size, the voxel size used in the developed compression scheme should be adjustable to match the size used in the application such that the voxel grids on both the sensor end and the application overlap. The area to be covered by 3D sensors is an industrial robotic cell, approximately 10 m wide, 10 m long and 4 m high. This requires multiple 3D sensors distributed around the cell, focusing inwards.

The end application does not require the RGB data that is generated by the Kinect sensors, thus only the depth measurements and the corresponding point clouds are of interest. However, as mentioned in the introduction, a single computer has limited USB bandwidth, which makes it practically impossible to connect all sensors directly to the central computer. By introducing an embedded computer placed at the sensor location, preprocessing and compression can be done locally before transferring the data to the central computer over a Gigabit local area network, as illustrated in Figure 1.

The software running on the embedded computer should be easy to deploy to an arbitrary number of sensor nodes from a remote location, i.e., it should not be necessary to compile or set up different versions of the software for nodes that use identical hardware. The system should also be scalable to such an extent that adding several more sensor nodes should not exhaust either the Gigabit Ethernet bandwidth or the CPU and GPU processing capabilities of the central computer. Thus, as much processing as possible should be performed locally at the sensor node, and efficient 3D point cloud processing and compression schemes are therefore needed at the embedded computer.

### 2.2. Point Cloud Preprocessing

The ROS node running on the sensor node is developed to process point clouds from different types of time-of-flight sensors. The software is designed to receive a “PointCloud2” ROS topic which can be generated by different ROS drivers depending on the sensor brand and type. Different point types can also be used, e.g., XYZ (coordinates only), XYZI (coordinates + intensity) and XYZRGB (coordinates + color). When a point cloud is received by the ROS node, it is first converted to an XYZI type cloud regardless of the input type. If the original point cloud already contains an intensity value, it is passed through as is (with some scaling), but, if not, a new intensity value is calculated. The generation of this intensity value will be described in detail in Section 2.5. Any RGB color information is discarded.

To minimize the amount of data which is transmitted on the network between the embedded and the central computer, the captured point cloud is then transformed into a global coordinate system and cropped. The transformation matrix from the sensor’s local coordinate system to the global coordinate system is known and published as an ROS topic by the central computer. By subscribing to this topic, the ROS node performs the transformation by using functions from the Point Cloud Library (PCL, [14]). Even though the sensors are statically mounted, the transformation matrix is looked up every time a point cloud is received, to ensure that any changes due to updated sensor calibration are incorporated. If, for some reason, the matrix is not received, the latest known transformation is used.

When the transformation is complete, most of the unwanted sensor data (i.e., data which lie outside of the robotic cell) can be filtered out using a simple box crop filter. The resulting point cloud now only contains points in the global coordinate system which are of interest to the application on the central computer. As both the transformation matrices for all sensors and the robotic cell size are known, the same method can be applied on an arbitrary number of sensor nodes without any need for customization. Transforming all points to the global coordinate system before compression also ensures that the voxel grid of the octree used by the compressor will overlap with the grid of the GPU Voxels map used by the central computer.

### 2.3. Data Representation

The resulting point cloud format after processing on the embedded computer was preferred to be voxel- or grid-based, as the application running on the central computer uses this kind of representation. By building on the result found by [2], the ROS node performs octree compression and encoding on the preprocessed point cloud stream before publishing the encoded stream to the network.

An octree is defined as a tree structure, where each node has zero or eight children. A point cloud can be inserted into an octree structure by encapsulating the whole volume in a bounding box with equal side lengths. The bounding box is then divided into eight subvolumes (children) called voxels, and the process is repeated for each subvolume that contains points. A fixed octree resolution $d_{v}$ can be selected such that the division process stops when the subvolumes reach a given size. The result is an octree, where all points of the cloud is encapsulated by the leaf nodes (the smallest voxels). Based on the resolution, each leaf node may encapsulate one or many points. The octree can be serialized to a binary sequence describing its structure, as shown in Figure 2.

The Point Cloud Library supplies several types of octrees and octree leaf nodes, including base classes. One of them is the double-buffered octree with point detail encoding, as outlined in [2]. To achieve the best possible compression ratios, the double-buffered octree was selected as the base octree type, and, by adapting the point detail encoding scheme, a new octree class was created. The encoding of the full point details was removed, and a new value of intensity was introduced. By creating a new leaf node type, each leaf in the octree is now capable of storing a floating point intensity value between 0 and 1. The creation of the intensity value will be described in Section 2.5. When the point cloud is inserted into the octree, the intensity value from all XYZI points that fall within each leaf node are accumulated. Thus, each voxel in the octree gets a value of trust or intensity based on the intensity values of all points in the original point cloud that was covered by the voxel.

### 2.4. Compression

When the ROS node is started, an octree resolution $d_{v}$ must be specified. The resolution of the octree is then kept constant as long as the process is running. This ensures that the tree depth is constant and that subsequent octrees can be compared using the XOR-method incorporated in the double-buffered octree. After the point cloud has been inserted into the octree, the tree structure is serialized to its binary sequence. For each leaf node, the new floating point intensity value is encoded as a separate 8-bit integer value, i.e., the floating point range 0–1 is converted into the integer range 0–255, which results in a maximum quantization error of ±0.00195. Using a byte to represent a floating point value is done to minimize the size of the compressed point cloud and, as will be shown in Section 2.5, the resolution is sufficient for the application at hand.

The entropy coder exploiting the double-buffered octree structure as well as the octree serialization function were used in their original and unmodified versions. An overview of the modified compression scheme is shown in Figure 3. The point detail encoding implemented in the original method generated sets of integer *x*, *y* and *z* coordinates relative to the leaf node origin for each point encapsulated in the voxel. The precision of the encoded points was controlled by limiting the range of these integer values, e.g., when using an octree resolution of 9 mm, a range of $\left[0,8\right]$ results in a precision of 1 mm. Limiting the range of the symbol set in this way allowed the point details to be efficiently encoded by the range encoder [2].

In our work, the point detail encoding was replaced by the generated intensity value stored in a single byte with a range 0–255. The range is not further limited, but, in contrast to the original method, our method guarantees that there will only be one such number for each voxel. Depending on the octree resolution and the sensor resolution, there could possibly be a large number of points encapsulated by each voxel. The original method requires that, for each point, the point details must be encoded, but when using an intensity value, only a single byte is needed. Both the serialized octree and the intensity values are finally passed to the range encoder, which results in a compressed byte stream that is published as an ROS topic.

### 2.5. Voxel Intensity Value Computation

Some versions of time-of-flight sensors natively provide an intensity value for each measured point. One example of such a sensor type is a lidar, where each measurement contains an intensity value based on the strength of the reflected signal. For other sensor types, e.g., RGB-D sensors such as the Kinect, such a value does not exist, and, when considering individual measurements, it is difficult to see how such a value could be constructed. However, when the point cloud has been converted to the “voxel domain” after being inserted into an octree, each voxel can contain several measurements, and that number of measurements can be seen as a measurement of trust in or intensity of the voxel.

The following paragraphs will describe different methods to construct the intensity value of a voxel, based on the measurements from a single Kinect sensor. One reason for introducing such a value is that, when using e.g., OctoMap or GPU Voxels in a later stage to fuse data from multiple sensors into a single map, the intensity values, if comparable and based on the same calculations, can be easily fused to create a level of trust for each voxel in the map. The intensity values can also be used to filter out voxels with weak measurements (typically noisy outlier points). Thus, the goal is to create an intensity value that makes sense in such scenarios.

#### 2.5.1. Counting Points

When constructing a voxel intensity value, perhaps the most intuitive method is to simply count the number of points that reside in each voxel such that the intensity value becomes $I_{v}=n$, where *v* is the considered voxel and *n* is the number of points inside the voxel. This gives a value of strength directly based on the number of measurements, where each point is given a point intensity
(1)$$I_{p}=1.$$
While this is a simple solution, it introduces an issue when the intention is to filter out noise from RGB-D sensors such as the Kinect V2 due to the fact that the measurement points are not evenly distributed throughout the volume of interest.

#### 2.5.2. Point Value Based on Quadratic Distance

The Kinect V2 has a field of view (FOV) of 70.6×60 degrees, and a resolution of 512×424 pixels for the generated depth map (a total of 217,088 pixels). As the octree resolution is constant, using the counting method to create voxel intensities means that voxels closer to the sensor will contain a much higher number of measurements than voxels farther away, when measuring the same object at different distances. Even a few noisy points (or false measurements) close to the sensor is enough to generate a much higher voxel intensity than accurate measurements farther away. Consider, for example, rain drops when using the sensors outdoors. The drops are evenly distributed in the entire volume but will trigger much higher intensity values close to the sensor, as they are “hit” by more measurements. To compensate for this behavior, a new method for generating intensity values is suggested.

Let *x* and *y* be the side lengths of the rectangle that is generated by intersecting the FOV with a plane perpendicular to the *Z*-axis (the depth axis) of the sensor (see Figure 4). Using the FOV angles, the lengths at a given depth *z* can be calculated as
(2)$$x\left(z\right)=2\;\cdot \;z\;\cdot \;\tan\left(\frac{35.3\;\cdot \;\pi }{180}\right),\;y\left(z\right)=2\;\cdot \;z\;\cdot \;\tan\left(\frac{30\;\cdot \;\pi }{180}\right).$$

Now, let $d_{x}\left(z\right)$ and $d_{y}\left(z\right)$ be the side length of each pixel in the depth map projected onto the plane. The area of each pixel for the Kinect V2 can be written as
(3)$$A_{p}\left(z\right)=d_{x}\left(z\right)\;\cdot \;d_{y}\left(z\right)=\frac{x(z)}{512}\;\cdot \;\frac{y(z)}{424}.$$

Let $d_{v}$ be the octree resolution, i.e., the side length of the cube forming the voxel, and let $d_{v}^{2}$ be the area of the voxel when projected onto the plane. The maximum number of measurements (pixels) that can be encapsulated by a single voxel at a given distance can then be found by
(4)$$N_{p}\left(z\right)=\frac{d_{v}^{2}}{A_{p}\left(z\right)}=\frac{d_{v}^{2}}{z^{2}\;\cdot \;c},\;$$
where *c* is constant for a given sensor type. It should be noted that *c* will be different depending on the sensor type and its parameters, i.e., the FOV and the resolution. Changing any of these values will impact the calculation of the point intensity value $I_{p}$. A finer resolution (more pixels in the same FOV), will result in a lower $I_{p}$, as more points will fit into a voxel. A larger FOV with the same resolution will have the opposite effect. In addition, note that $N_{p}\left(z\right)$ is an approximation: The voxels have a fixed placement in the global coordinate system and, depending on the transformation between the sensor and the global coordinate system and the distance *z*, $d_{v}$ may not accurately describe the area of the voxel when projected onto the plane, since the voxel may be rotated or offset relative to the plane. However, since the calculation of the point intensity value is performed before the points are transformed, this approximation is not practically avoidable. The point intensity value $P_{v}$ for a given measurement point can now be defined as
(5)$$I_{p}\left(z\right)=\frac{1}{N_{p}\left(z\right)}$$
such that each point is given an intensity value inversely proportional to the maximum number of points that could fit inside the voxel. This suggested method for calculating the voxel intensity place low value in measurements close to the sensor, but, when looking at it from the voxel perspective, all voxels will have comparable intensities, and as the experimental results will show, when using the generated value to filter out “weak” voxels, the ones that are removed will be distributed over the entire volume, as opposed to the counting method where voxels closer to the sensor are prioritized, and accurate measurements far from the sensor are removed more quickly.

#### 2.5.3. Point Value Based on Linear Distance

A third method for generating the voxel intensity is to use a slightly less aggressive approach when lowering the point intensities close to the sensor, so that voxels with multiple measurements are awarded a higher intensity value. To achieve this, the cutoff distance $z_{c}$ at which it is impossible to get more than one measurement inside a voxel is calculated, i.e., where $N_{p}\left(z\right)=1$:(6)$$z_{c}=\sqrt{\frac{d_{v}^{2}}{c}}.$$
When inserting an octree resolution $d_{v}$ of e.g., 0.04
m, this distance is 14.57 m. The point intensity value is then defined as simply
(7)$$I_{p}\left(z\right)=\frac{z}{z_{c}}.$$

Figure 5 shows $I_{p}\left(z\right)$ for the three different methods outlined above for the Kinect V2 depth sensor. The values are capped at 1.0 after the cutoff distance $z_{c}$, thus preventing single point intensities from exceeding the 0–1 range.

#### 2.5.4. Voxel Intensity

In the previous paragraphs, three methods for generating point intensity were presented. The final step is to generate an intensity value for the voxel that is encapsulating the points. The voxel intensity is defined as follows:(8)$$I_{v}=\sum_{i=1}^{n}I_{p}\left(i\right),$$
where *n* is the number of points inside the voxel. When using $I_{p}$ from Equation (Equation 1), the generated value is an integer value corresponding to the number of points. Using Equation (Equation 5), $I_{v}$ will accumulate towards 1 when the voxel is filled up with points. However, as $N_{p}\left(z\right)$ is an approximation, $I_{v}$ may accumulate to slightly more than 1, hence $I_{v}$ is capped in the software so that it will never exceed 1 when using Equation (Equation 5). The voxel intensity generated based on Equation (Equation 7) may accumulate to more than 1 if there are multiple measurements inside the voxel. This must be taken into account during the octree serialization process, where the intensity value is converted to an 8-bit integer value by the following method: (9)$$I_{v}^{int}=\left\{\begin{array}{cc} I_{v}\;\cdot \;255 & \mathrm{for}\;I_{v}\leq 1,\\ 255 & \mathrm{for}\;I_{v}>1. \end{array}\right.$$

As described in Section 2.4, the maximum quantization error when converting a floating point range 0–1 into an integer range 0–255 is ±0.00195. If only a single point is registered in a voxel, the point intensity value $I_{p}$ must be larger than the quantization error for it not to be set to zero. For $I_{p}$ according to Equation (Equation 1), this is not an issue as the value is already an integer and does not need to be converted. This means that, if there are more than 255 points in a voxel, the intensity value will be capped at 255 and not reflect the real point count. However, this can only happen when the Kinect V2 measures objects closer than approximately 1 m when using a 4 cm octree resolution, which will never happen in our scenario.

The worst case scenario happens when using Equation (Equation 5), where point intensities close to the sensor are very small. However, it can be shown that, when using an octree resolution of 0.04
m, which is the largest resolution used in the experiments in this paper, the smallest possible point intensity is 0.0047 at a distance 1.0
m from the sensor, which is more than double the quantization error.

The methods used in the examples in this section are only valid for the Kinect V2 or similar sensors. Other methods for generating the voxel intensity may be designed for different sensor types or depending on the application. For example, for the Velodyne PUCK lidar connected to one of the sensor nodes, the intensity value already exists and does not need to be calculated.

### 2.6. Decompression and Denoising

The ROS node which runs on the central computer subscribes to, decodes and decompresses the encoded stream into reconstructed point clouds as shown in Figure 6. After the compressed stream has passed through the entropy decoder, the output point cloud is generated by combining the deserialized octree structure and the voxel intensity values. As mentioned in Section 2.2, the octree resolution should be selected to match the requirement of the end application. In our scenario, the application is intended to be based on the GPU Voxels library. By matching the size of the leaf nodes in the octree used by the compressor with the resolution of the GPU Voxels map, the reconstructed point cloud can be inserted with a one-to-one ratio. To achieve this, the center coordinate of the leaf voxels needs to be reconstructed by the decompressor.

For each leaf node in the received octree, a point is generated from the voxel center coordinate and the intensity value of the voxel. The result is a new point cloud including coordinates with a precision equal to the octree resolution, similar to a voxel grid filter. Every leaf node of the decompressed octree thus results in an XYZI point in the new point cloud, including the voxel intensity value in the range 0–255 (see Figure 7).

After decompression, the generated intensity value can be further exploited. By using a pass-through filter, the point cloud can be filtered such that points with an intensity below a given value are removed.

### 2.7. Experimental Setup

The point cloud processing and compression scheme described in this paper is intended for use in a large scale industrial location. The system was therefore developed and tested in an indoor robotic cell consisting of two rail-mounted ABB IRB4400 robots (ABB Ltd., Zurich, Switzerland) and one gantry-mounted ABB IRB2400 robot (see Figure 8). In addition to the robots, a processing facility was placed in the cell to introduce a more realistic environment.

The area to be covered by 3D sensors is approximately 10 m wide, 10 m long and 4 m high. To accomplish this, six Kinect V2 sensors were mounted at different locations along the walls at a height of around 4.2 m. In addition, a single Velodyne PUCK VLP-16 lidar (Velodyne LIDAR, San Jose, CA, USA) was mounted in one of the corner locations.

The ROS nodes for preprocessing and compression were deployed on six NVIDIA Jetson TX2 Development Boards, each connected to their own Kinect V2 depth sensor. one node is also connected to a Velodyne VLP-16 PUCK lidar and, for future use, a Carnegie Robotics Multisense S21 stereo camera (Carnegie Robotics LLC, Pittsburgh, PA, USA), as seen in Figure 9. The Jetson TX2 contains a Quad-core Arm A57 CPU (Arm Limited, Cambridge, UK), an NVIDIA Pascal GPGPU and 8 GB LPDDR4 memory. The ROS node for decompression and denoising was deployed on the central computer, which was equipped with an Intel Core i5-2500 CPU (Intel Corporation, Santa Clara, CA, USA) and 8 GB of system memory.

By equipping sensor nodes with different sensor technology and enclosing the electronics in a waterproof cabinet, the goal is to use the sensors in an outdoor environment, where measurements from the different sensors can compliment each other in different weather and lighting conditions. In fact, the sensor nodes have already been tested outdoors in an industrial area, and evaluating the results from these tests is part of our future work.

To generate the point cloud stream which is input to the compressor, an ROS driver for the Kinect V2 (IAI Kinect2, [15]) was installed. This driver generates depth and color images at a rate of 30 frames per second, which in turn are converted and published as ROS PointCloud2 messages. The driver uses the NVIDIA GPU for depth image processing. Figure 9c shows a block diagram of the hardware and software used in the experimental setup.

In the experimental results presented in Section 3.2 and Section 3.3, the compression and decompression processes from a single sensor node to the central computer was considered. Only the Kinect V2 data was processed, and the point counting method for generating voxel intensities was used.

The transformation from the sensor’s coordinate system to the global coordinate system was manually measured and has an unknown accuracy. However, the transformation only affects which points are removed by the crop-filter, and thus does not directly affect the compression performance. All experiments were conducted using live point cloud streams.

### 2.8. Multisensor Setup

For the denoising results presented in Section 3.4, all six sensor nodes were used. The sensors were placed along the outer peripheral of the lab and manually calibrated based on the method in [16]. Calibration of the camera intrinsic parameters was performed according to [17]. Note that, compared to the setup in [16], some of the sensors’ mounting locations have been moved such that the monitored area is smaller. Work has also been done to create an automatic calibration scheme for the same industrial lab. These results have been submitted and are currently under review in [18]. Table 1 shows the locations of the sensors in the global coordinate system.

Mounting several Kinect sensors in the same environment could possibly lead to interference problems. Indeed, the first version of the Kinect is known to have problems with interference, as they are based on structured light [19]. The latest version that we used in our experiments (Kinect for XBOX One) is based on time-of-flight measurements using a modulated continuous wave IR signal. There is still a possibility of interference when using this version of the sensor, but it has been shown that the errors caused by interference are negligible when certain mounting constellations are avoided [20]. Even though some of the sensor orientations in our lab may fall within this “bad” configuration, we have not had any visible problems with interference in our experiments, and thus no further steps have been taken to manage this potential issue.

## 3. Results

The resolution of the Kinect V2 IR depth image is 512 × 424 pixels, thus the number of points in each point cloud generated by the IAI Kinect2 ROS driver is 217,088. When discarding the RGB color information which is not used, each point consists of 4 32 bit floating point numbers, or 16 bytes. (The PCL point type used includes *x*, *y*, *z* coordinates and four bytes of padding). These numbers are constant for all point clouds and give an original point cloud size of 3392 KiB.

### 3.1. Preprocessing

The first step in preprocessing was to generate the point intensity values for all points. In this experiment, Equation (Equation 1) was used. In the process, the points were converted from XYZ to XYZI (this does not increase the point cloud size due to the four bytes of padding in the XYZ type point). Before the point cloud was sent through the octree compressor, it was transformed and crop-filtered as previously described. This yielded a new point cloud with an average of 37,108 points when measuring 1000 consecutive point cloud frames, which was then sent through the compressor at the sensor node. Table 2 shows the experimental results from this part of the process. By cropping the point cloud, the size was reduced by an average factor of 5.85.

### 3.2. Compression

After preprocessing, the point cloud was processed by the compressor. When inserting them into an octree structure with a 4 cm octree resolution, the same 1000 point clouds resulted in octrees with an average of 17,771 octree leaf nodes each. This means that, on average, 2.09 points were inserted into each voxel. The compression ratios and sizes were logged by the compressor software. The number of bytes in the transferred encoded stream was counted each frame to measure the exact size of each compressed point cloud. Table 3 shows the experimental results from the compression process. A compression ratio (based on size) of 40.5 was achieved.

Another experiment was done with an octree resolution of 2 cm. This was performed on another point cloud stream, hence the size of the cropped point cloud is slightly different. The results from this experiment can be seen in Table 4. A compression ratio of 22.5 was obtained using 2 cm octree resolution.

The number of bytes per point in both experiments was reduced by a factor of 20 from 16 to 0.82 and 0.8, respectively. These results indicate that the encoding process presented by [2] is providing similar performance in both experiments. The larger compressed size and lower compression ratio in the second case are due to a higher number of octree leaf nodes as a result of a more fine-grained resolution.

The IAI Kinect2 driver can deliver compressed depth images, which can limit the bandwidth required to transfer depth data. The depth images could be transferred to the central computer before being converted to point clouds. Measuring 1000 depth images from the same sensor node yields an average depth image size of 0.43 MB, and a compressed image size of 0.22 MB when using the default JPEG compression. Compared to transferring the depth data in a raw point cloud format, using the compressed depth image would yield better results (the cropped, uncompressed point cloud has a size of approximately 0.58 MB as seen in Table 3). However, the octree compression method far outperforms the depth image compression, yielding compressed clouds of only 14.31 KiB and 26.06 KiB when using an octree resolution of 4 and 2 cm, respectively.

### 3.3. Frequency and Bandwidth

Table 5 shows the results obtained by logging the application and system performance. The frequency of the transferred point clouds was measured using the built-in ROS topic monitor, and the cycle time of the software was measured using the rostime C++ library. CPU load and memory usage were not formally measured but estimated based on the Linux process monitoring utility htop.

The maximum frequency and bandwidth of the compressor were measured while compressing point clouds using a 4 cm octree resolution. While the IAI Kinect2 ROS driver is able to deliver point clouds at a rate of 30 per second, the compression process is currently limited by running on a single CPU core. The developed ROS node was able to process, compress and transfer point clouds at an average rate of 26.9 Hz measured over a window of 10,000 frames. This results in an average bandwidth of 384.9 KiB/s, based on the compressed size in Table 3.

By logging the processing time for the ROS node function responsible for the preprocessing and compression processes, an average time of 36 ms per frame was measured over 1000 frames. This corresponds to a maximum possible frame rate of 27.8 Hz, which is in accordance with the above results when taking into account that there is some additional overhead caused by the underlying ROS system and that the process used approximately 100% of a single CPU core.

When limiting the IAI Kinect2 driver to output point clouds at 20.0 Hz, the developed ROS node is able to process and compress all incoming point clouds, resulting in a bandwidth of 286.2 KiB/s. At this configuration, the ROS node utilizes around 80% of the processing capability of one CPU core on the Jetson TX2. In addition, the node used approximately 60 MiB of memory.

On the central computer, the average amount of time used to decompress a single frame, based on 1000 measurements, was 13 ms, as seen in Table 5. The process utilized around 14% of a single CPU core and 35.7 MiB of memory. As a final test, point clouds from all six sensor nodes were decompressed simultaneously. This utilized around 65% of the CPU core and 51.3 MiB of memory, which indicates that there is some overhead in the ROS software and that the actual resources needed for decompression are less than the results presented in Table 5.

Figure 10 shows a visual comparison between uncompressed and compressed point clouds. Figure 10a,c show the cropped point cloud with RGB color, where Figure 10c is accumulated over one second for better visibility. Figure 10b,d show the same region of interest after compression and decompression, using a 4 cm octree resolution. The decompressed clouds were filtered using the calculated intensity value, such that points with an intensity value lower than 2 were removed. To highlight the underlying octree structure, the points are displayed using cubes.

### 3.4. Denoising

When the compressed point cloud has been reconstructed by the decompressor, the new intensity value can be used to filter the points, as suggested in Section 2.5. To test the feasibility of this suggestion, point clouds from all six sensor nodes, including the Velodyne lidar, were compressed, transmitted, reconstructed and filtered using different intensity values. Any formal verification of the results has not been performed, but the results were visually inspected using a visualizer application from ROS. Figure 11 presents the visualized results, when using the different point intensity generation algorithms and filtering using different values. A stronger color corresponds to a higher intensity. Points with an intensity value lower than the filter value are removed. As seen in Figure 11a,c, there is a relatively large amount of noise in the point clouds, especially close to the sensor origins. In the filtered clouds, the floor and points outside the cell have been removed by the box crop filter.

Filtering the points on the “point count” intensity (Figure 11d) does a decent job of cleaning up the cloud. However, as can be seen from the picture, some noisy points close to the sensors are still present, and points further away are aggressively removed. This is best seen from the yellow points in the bottom left corner, which originates from the sensor placed in the top right corner (the origin of the sensor lies just outside of the image, refer to Figure 11b for the sensor locations).

Using the “linear” intensity value, all noisy points close to the sensor origin are removed, and more of the measurements further away are preserved, as seen in Figure 11e. In the authors’ opinion, this is the point cloud that best represents the measured environment, with the least amount of noise and strong measurements of the actual objects.

Figure 11f shows the points filtered using the “quadratic” point intensity value. Here, it is seen that points closer to the sensor (e.g., on the blue column on the left) are given a relatively lower intensity value, and points farther from the sensor (e.g., the yellow points in the lower bottom corner and close to the floor) are given a relatively higher intensity value. The effect is, as expected, that the points removed by the filter are evenly distributed throughout the volume, and not based on the distance to the sensor.

In all the filtered clouds, almost all points generated by the Velodyne lidar (best seen as the olive-colored points on the right wall) are conserved, due to the fact that the intensity is generated by the sensor itself and not calculated, and that these values are higher than the filter values. This behavior is intended, as the Velodyne measurements are much more stable and reliable than the Kinect measurements, and thus should not be filtered out as noise. In the examples, the filter values were picked by trial and error, and different values would show different results (i.e., more points removed by using a higher filter value). The values should therefore be tuned to the application requirements and the user’s needs.

## 4. Discussion

In this paper, a scalable solution for 3D sensing in a large volume consisting of multiple sensors was presented. A constant frame rate of 20 Hz was achieved on the sensor nodes containing an embedded processing unit. The bandwidth requirement on each local node was 286.2 KiB/s, which means that the proposed solution could be scaled up to an order of 440 sensors (assuming that the central node has a dedicated 1 Gbit/s network input and “unlimited” processing power). Compared to transferring the cropped, but uncompressed point cloud, which would result in a bandwidth of 11596 KiB/s ( 579.8 KiB × 20 Hz), the compression leaves room for significantly more point clouds streams on the network. It was also shown that the octree compression outperforms the default depth image compression performed by the IAI Kinect2 driver.

In [2], the number of bytes per point (BPP) when using an octree resolution of 9 mm and a fixed point precision of 9 mm was 0.30. When using a point precision of 1 mm, the BPP was 0.87, which is closer to our results. The lower values achieved in [2] is most likely due to the fact that the range of the symbol set used to encode the point details is greatly limited (the range $\left[0,8\right]$ was used), which makes the range encoder more efficient. The results also show that, while in the original method the BPP increases with finer precision, in our method, the BPP is practically unaffected when the octree resolution is changed.

It should also be noted that since point details for all points are encoded in the original method, the total compressed size of the point cloud would be larger. Even with a BPP of 0.3, the resulting compressed cloud would have a size of 63.6 KiB, corresponding to a bandwidth of 1272 KiB/s. Thus, when comparing the results, one should also consider the application at hand. In our scenario, it was more important to lower the size of the compressed point cloud and to generate the intensity values which are to be used by the end application than to encode point details for all points in the original point cloud.

In [4], a frame rate of 5.86 Hz was achieved with a powerful CPU (Intel i7), while 20 Hz was achieved in this paper using only a much less powerful ARM processor. The reason for the improved performance achieved in our work results from (1) a dedicated local network with no background transmission and (2) all the data points inside one voxel are in our work filtered and described by only one coordinate and one intensity value.

Different algorithms for generating voxel intensity values based on measurements where no such value exists were proposed. The experiments showed promising results when using the generated values to filter out false or noisy measurements. By adapting the algorithms, this method can be used to generate intensity values for different sensor types, not only RGB-D sensors as described in this paper.

Future work includes evaluating the benefit of filtering based on intensity values on data from outdoor testing (e.g., removing noise from rain and other unwanted reflections). In the future, measurements from the Carnegie Robotics stereo camera will also be included. In addition, an effort will be made to optimize the software. More specifically, the possibility of limiting the symbol range used when encoding the intensity value should be explored, as this could make the range compressor more efficient. Parts of the process can also be parallelized in order to utilize multiple CPU or even GPU cores on both the Jetson TX2 module and the central computer. This could make it possible to compress streams at higher frame rates and reduce latency.

## Figures and Tables

**Figure 1 sensors-19-00636-f001:**
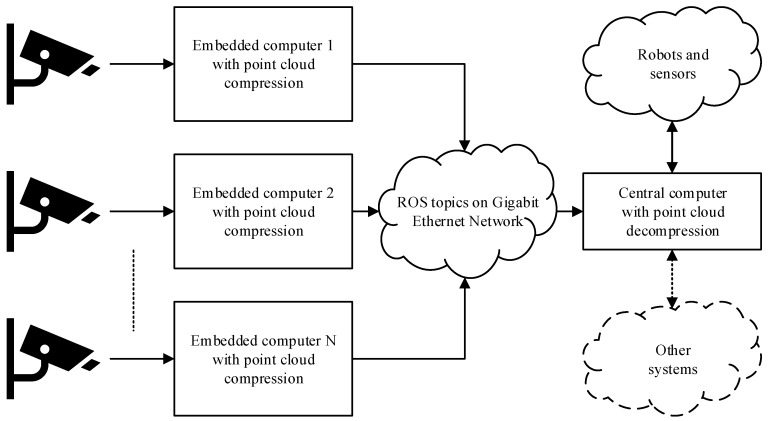
Schematic overview of the considered sensor network. The number of sensors and embedded computers is scalable.

**Figure 2 sensors-19-00636-f002:**
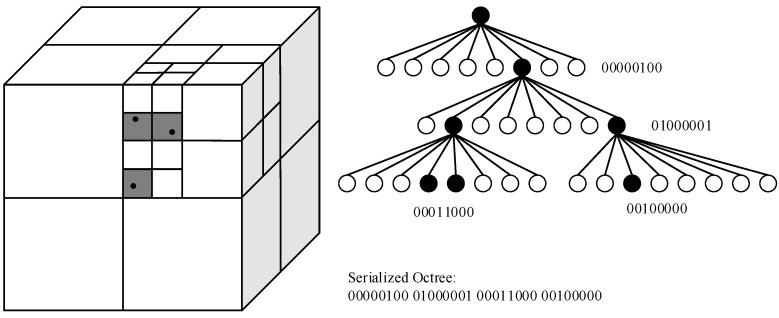
Schematic overview of the octree data structure and its serialization. Nodes in the tree that encapsulate points are marked as occupied (binary 1). These are the only nodes that have children. On the finest level, the division process is stopped, and the nodes that encapsulate points are marked as occupied. Figure ©2012 IEEE. Reprinted, with permission, from [2].

**Figure 3 sensors-19-00636-f003:**
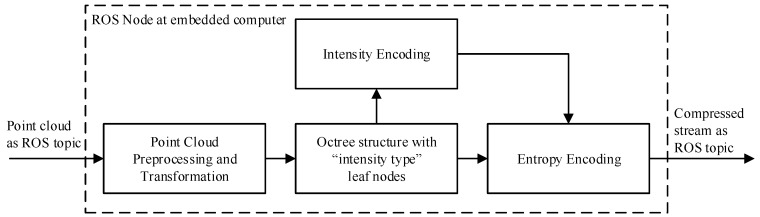
Schematic overview of the compression principle implemented as an ROS node at the embedded computer.

**Figure 4 sensors-19-00636-f004:**
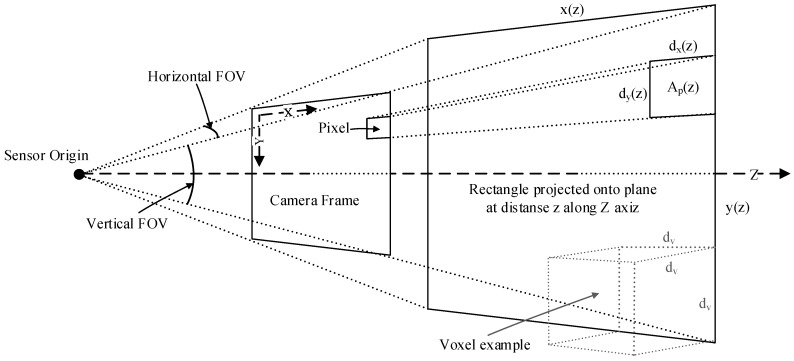
Schematic view of a sensor’s field of view when projected onto a plane perpendicular to its direction of view.

**Figure 5 sensors-19-00636-f005:**
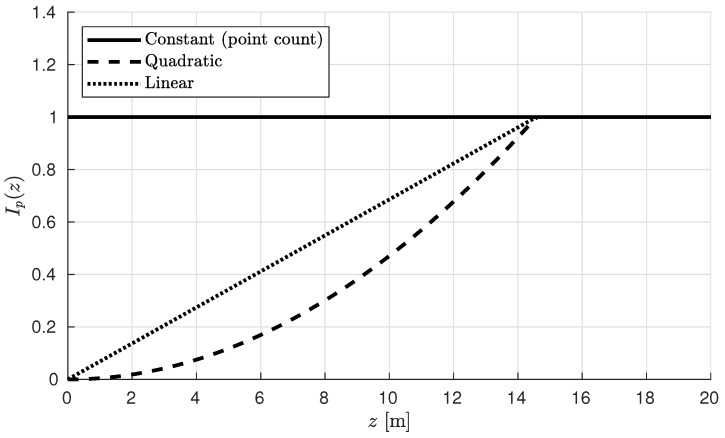
Plot of the point intensities $I_{p}\left(z\right)$ for the three discussed calculation methods.

**Figure 6 sensors-19-00636-f006:**
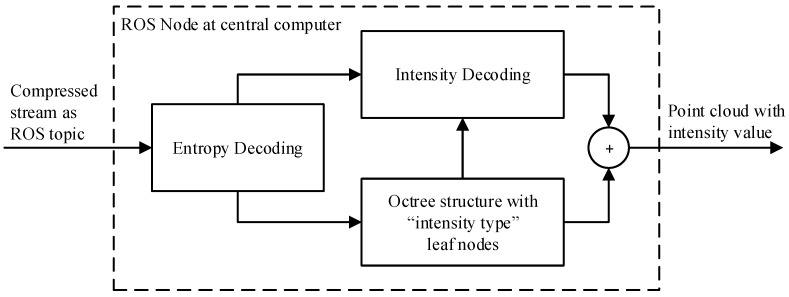
Schematic overview of the decompression principle implemented as an ROS node at the central computer.

**Figure 7 sensors-19-00636-f007:**
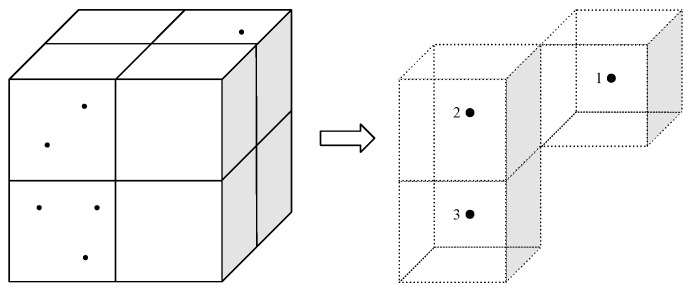
(**Left**) The points of the original point cloud inserted into an octree structure. The figure shows only a subset of the octree consisting of eight leaf nodes, where three of the voxels are occupied. (**Right**) After decompression, the reconstructed point cloud contains points at the center coordinate of the occupied voxels, in addition to the intensity value. In this example, the intensity value is generated by point counting. The dashed voxel cubes on the right-hand side are only shown for reference.

**Figure 8 sensors-19-00636-f008:**
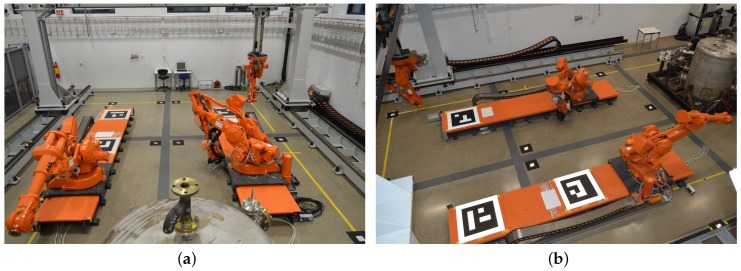
(**a**) overview of the robotic cell which is covered by the embedded sensor nodes; (**b**) the cell seen from a different angle, close to one of the sensor nodes.

**Figure 9 sensors-19-00636-f009:**
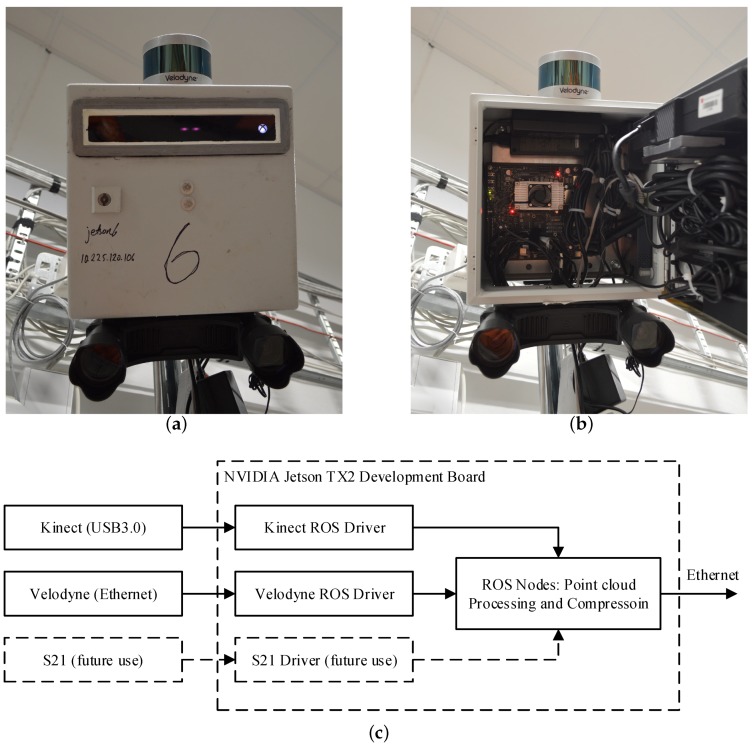
(**a**,**b**): embedded sensor node consisting of an NVIDIA Jetson TX2 Development Board, a Kinect V2 depth sensor, a Velodyne VLP-16 PUCK Lidar and a Carnegie Robotics S21 stereo camera for future experiments; (**c**) schematic overview of the NVIDIA Jetson TX2 hardware connections and software modules. The points from the Velodyne lidar and the S21 stereo camera are not part of the experimental results.

**Figure 10 sensors-19-00636-f010:**
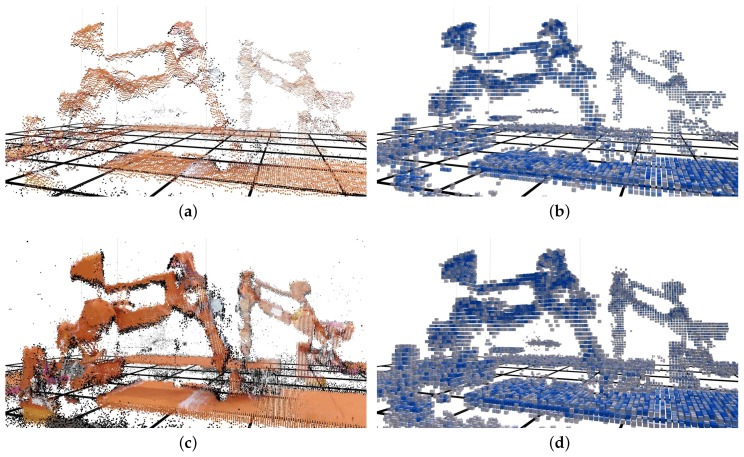
Visual comparison of original and compressed point clouds using a 4 cm octree resolution. (**a**,**c**): colored point cloud generated by the Kinect V2 sensor. (**b**,**d**): point cloud generated by the decompressor at the central computer. For better visibility of the underlying octree structure, the points are shown as cubes with 3 cm sides. A stronger color indicates higher intensity.

**Figure 11 sensors-19-00636-f011:**
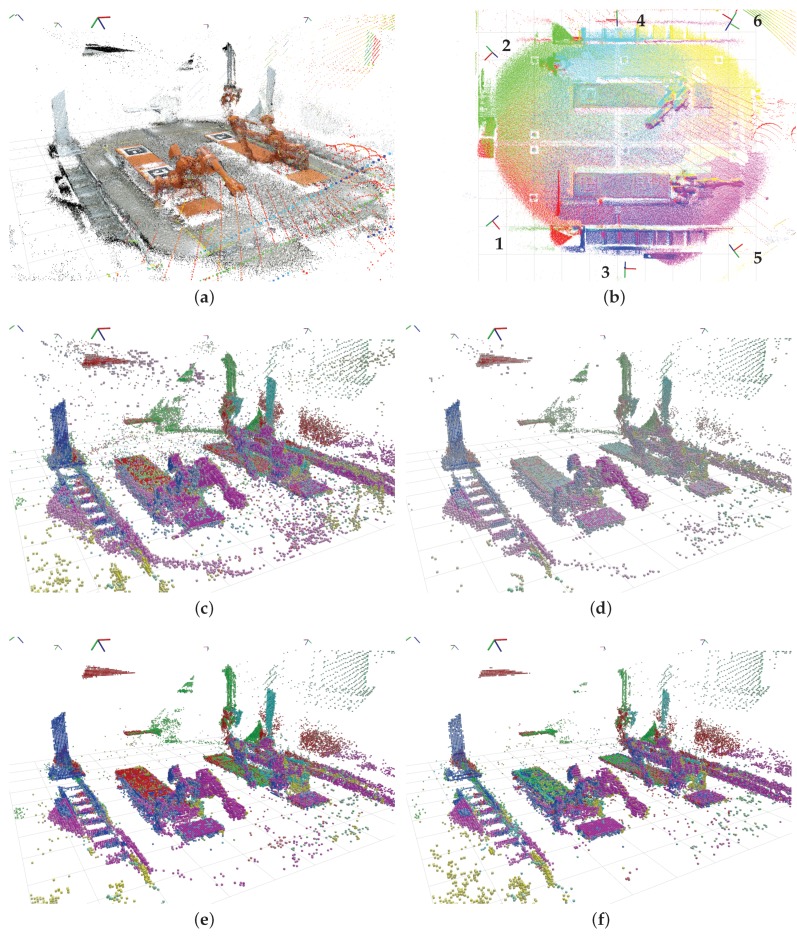
(**a**) original point clouds from all sensors, including color information for the Kinect sensors; (**b**) top down view of original point clouds, where the different sensors’ points are color coded; (**c**) reconstructed, unfiltered point cloud; (**d**) cloud with “point count” voxel intensities, filter value 2; (**e**) cloud with “linear” intensities, filter value 64; (**f**) cloud with “quadratic” intensities, filter level 64.

**Table 1 sensors-19-00636-t001:** Calibrated sensor positions in meters and orientations in degrees. The positions are in the global coordinate system. *N* corresponds to the sensor number.

*N*	X	Y	Z	RotZ	RotY	RotX
1	7.798	0.496	4.175	40.170	0.639	−136.170
2	1.729	0.501	4.135	−45.253	2.063	−141.490
3	9.522	5.275	4.353	88.439	−0.072	−143.461
4	0.553	4.995	4.353	−89.208	−0.495	−145.591
5	8.879	9.192	4.194	126.733	−1.311	−139.272
6	0.559	9.136	4.145	−121.458	−0.487	−137.562

**Table 2 sensors-19-00636-t002:** Point cloud filtering and cropping results. Cropped results are mean values over 1000 measured point clouds.

Measurement	Original	Cropped
Number of Points	217,088	37,108 ± 293
Size (KiB)	3392	579.8 ± 4.6
Ratio	1:1	1:5.85 ± 0.05

**Table 3 sensors-19-00636-t003:** Octree Compression Results, 4 cm octree resolution. The results are mean values over 1000 measured point clouds, with the number of points rounded to the nearest integer.

Measurement	Cropped	Compressed
Number of Points	37,108 ± 293	17,771 ± 118
Size (KiB)	579.8 ± 4.6	14.31 ± 0.20
Bytes per Point	16	0.82 ± 0.01
Compression Ratio	1:1	1:40.5 ± 0.5

**Table 4 sensors-19-00636-t004:** Octree compression results, 2 cm octree resolution. The results are mean values over 1000 measured point clouds, with the number of points rounded to the nearest integer.

Measurement	Cropped	Compressed
Number of Points	37,574 ± 313	33,321 ± 243
Size (KiB)	587.1 ± 0.49	26.08 ± 0.17
Bytes per Point	16	0.80 ± 0.01
Compression Ratio	1:1	1:22.5 ± 0.1

**Table 5 sensors-19-00636-t005:** Compression and decompression performance when using 4 cm octree resolution.

Measurement	Compression (@ max. FPS ${}^{1}$)	Compression (@ 20 Hz)	Decompression (@ 20 Hz)
FPS	26.9 HZ	20.0 HZ	20.0 HZ
Bandwidth	384.9 KiB/s	286.2 KiB/s	286.2 KiB/s
Cycle Time	(36±4) ms	(36±4) ms	(13±2) ms
CPU load	100%	80%	14%
Memory	63.5 MiB	60 MiB	35.7 MiB

${}^{1}$ Frames per second.

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
