# Peer review of "Embedded Processing and Compression of 3D Sensor Data for Large Scale Industrial Environments"

_sensors, 2019, doi:10.3390/s19030636_

Round 1
Reviewer 1 Report
This paper presents a scalable embedded solution for processing and transferring 3D point cloud data. The experiments shows that the proposed method has a good performance. The topic of this paper is important and meaningful. However,there are some aspects that should be improved. The main comments are as follows:
1. In the introduction, the subtitle is incomplete. For example, there is only 1.1 related works in "1. Introduction".
2. The contributions of the paper are not clear. I suggest authors should rewrite the organization of the introduction section.
3.The label of the formula is confusing.
4.The algorithm proposed in the paper has achievee good results in application, but it lacks theoretical deduction and proof.
5. What's the influence of the parameters in the algorithm? Such as c, dv,(4)e and so on.
6. What's the main difference of the proposed method with other compressing and transmitting data methods? The section 2 is not expressed the original method clearly.
7.In the experiment section, there is no comparsions with other methods, such as other preprocessing methods and compression methods. I think this is needed to evaluate your proposed method and prove the advantages.
8. How to evaluate the performance of compression and decompression? Is there a quantitative evaluation method?
Besides," Fig. 10 shows a visual comparison between uncompressed and compressed point clouds" and the "Denoising" section, there are only qualitative analysis in this paper. Quantitative analysis is not given to illustrate the difference of the compression and decompression for the proposed algorithm, which compares with the original data and other methods.
Author Response
Pleas see attached PDF document.
Yours sincerely,
Joacim Dybedal
Corresponding Author.

Reviewer 2 Report
It is very interesting and actual work on 3D range data sensors fusion and data processing. The proposed voxel type calculations are not new, but the possibility to fuse several sensor nodes and transmit data to central node is very interesting. Overall view of manuscript is very good, neglecting the fact that Microsoft has discontinued production of Kinect sensors (it was the cheapest and very reliable 3D sensor in a last 5 years). What I am missing in the text is: The way the sensors were placed in the environment 10x10x4? The way sensors were calibrated for this environment? The procedures on how the signal interference was managed for multiple Kinect sensors. And in my opinion, al formulas should be numbered. Good luck!
Author Response
Please see attached PDF document.
Yours sincerely,
Joacim Dybedal
Corresponding Author
